# Evolutionary Perspective of Nonclassical MHC Class I and Innate-like T Cells Relevance in Immune Surveillance

**DOI:** 10.3390/cells14201592

**Published:** 2025-10-14

**Authors:** Jacques Robert, Elnaz Najafi-Majd

**Affiliations:** Department of Microbiology and Immunology, University of Rochester Medical Center, Rochester, NY 14642, USA; elnaz.najafi.majd@gmail.com

**Keywords:** comparative immunology, MHC evolution, vertebrates, unconventional T cells

## Abstract

Unlike conventional T cells, which express a highly diverse repertoire of dimeric αβ T-cell receptors (TCRs) restricted by classical, polymorphic MHC class I molecules (MHC-Ia), a distinct group of T cells—collectively termed “innate-like T (iT) cells”—exhibits limited TCR diversity and depends instead on nonclassical, nonpolymorphic MHC class I molecules (MHC-Ib) for their development and function. While mounting evidence supports the role of iT cells as pivotal regulators and effectors in both innate and adaptive immune responses, many aspects of their biology remain incompletely understood. In humans, iT cells represent a significant fraction of the total T cell population, and evolutionarily conserved subsets have also been identified in other mammals and amphibians. Moreover, the expanding catalog of nonpolymorphic *MHC-Ib* genes and lineages—distinct from polymorphic *MHC-Ia* genes—across jawed vertebrate genomes suggests a broader and potentially more integral role for MHC-Ib molecules in T cell function and immune surveillance. In this review, we explore the immunological significance of MHC-Ib molecules and iT cells through an evolutionary lens, highlighting recent advances that shed light on their contributions to immune homeostasis and defense.

## 1. Introduction

Immunosurveillance refers to the process by which the immune system can detect not only microbial pathogens but also precancerous and cancerous cells in an organism (reviewed in [1,2]). Key components of the immunosurveillance process are classical major histocompatibility complex (MHC-Ia) molecules that are encoded by highly polymorphic genes and expressed at the surface of most cells in jawed vertebrates (reviewed in [3]). Over the past two decades, multiple tumor-specific and tumor-enriched peptides binding to MHC-I have been identified and shown to be able to activate anti-tumor CD8 cytotoxic T lymphocytes (CTLs). However, another group of T cells collectively termed “innate-like T (iT) cells” is increasingly recognized as an important player in immune surveillance. These iT cells exhibit features of both innate and adaptive immune characteristics and play a pivotal role in this process by functioning at the interface of the two immune systems. These iT cells appear to form a sophisticated defense network still not fully defined, which enables the organism to monitor and combat cancer effectively [2,4,5]. Indeed, iT cells can respond quickly to cellular stress or transformation. Critically, their functions often depend on nonclassical MHC class I molecules (MHC-Ib), which are more specialized and non-polymorphic than classical MHC class I. Some of these MHC-Ib molecules present unconventional antigens or serve as “distress signals” to the immune system, aiding in the identification of infected or transformed cells (reviewed in [6]).

This review explores the co-evolution of MHC-Ib and iT cells across jawed vertebrates—from cartilaginous and bony fish to amphibians, reptiles, birds, and mammals— and their contribution to immune surveillance. There are many excellent and comprehensive reviews available on mammalian MHC-Ib and innate T cells, especially CD1d/iNKT and MR1/MAIT systems. So, we have summarized this part on mammals and focused on nonmammalian systems. A comparative approach can shed some light on how the respective diversification and specialization of MHC-Ia/conventional T cells versus the MHC-Ib/iT cells have contributed to immune surveillance through evolution.

## 2. Classical Versus Nonclassical MHC-I

Classical MHC-I (*MHC-Ia*) genes encode transmembrane glycoproteins that play a crucial role in immune recognition (reviewed in [7]). These genes first arose in jawed vertebrates, and their fundamental role has been conserved throughout vertebrate evolution [3,8,9]. In all jawed vertebrates, *MHC-Ia* genes are highly polymorphic and located in the MHC genomic locus proper, which includes a plethora of genes (around 500) [10,11,12]. At the cell surface, MHC-Ia molecules present peptide fragments of eight to nine amino acids recognized via TCR of CD8^+^ T cells [13]. These molecules require association with β2-microglobulin and short peptides of seven to eight amino acids in the endoplasmic reticulum for cell surface expression. MHC-Ia downregulation is a well-known immune evasion method used by viruses and cancer. Loss of MHC-Ia expression is often correlated with faster disease progression, poorer survival, and more metastasis (reviewed in [14]).

In contrast to highly polymorphic *MHC-Ia* genes, nonclassical MHC-I (*MHC-Ib*) genes exhibit limited to no polymorphism and are often located outside the main MHC locus [15]. In eutherian (placental) mammals, *MHC-Ib* genes have typically more restricted patterns of expression. Notable examples in humans include *HLA-E*, *HLA-F*, and *HLA-G* and their homologs *Qa-1*, *Qa-2*, *HFE*, and *RT1* in mouse and rat, as well as the cluster of differentiation 1d (*CD1*) family and MHC-related protein 1 (*MR1*). Some of these MHC-Ib molecules are involved in immune regulation and stress surveillance. For example, HLA-E binds MHC-Ia leader peptides and interacts with NK cell receptors to inhibit NK cell attack on healthy cells. HLA-G, expressed at the maternal–fetal interface, plays a crucial role in promoting immune tolerance during pregnancy. HLA-F appears to regulate the immune system in pregnancy, infection, and autoimmunity by signaling through NK cell receptors. Notably, HLA-F can bind unusually large peptides, and can be expressed on the cell surface either associated with β2-microglobulin or as an empty monomer. Furthermore, it can interact with both TCR and the inhibitory LIR1 receptor [16]. Other MHC-Ib proteins including MIC, MILL, MR1, EPCR, and FcRN, as well as MHC-like MICA and MICB have been reviewed elsewhere [17].

In marsupials and monotremes, an extended family of 17 divergent *MHC-Ib* genes referred to as UT have been described [18]. Some *UT* gene family members are expressed in the thymus of the gray short-tailed opossum, notably *UT8* which is expressed by developing thymocytes [18,19]. Two *MHC-Ib* lineages have become of particular interest owing to their ability to present unconventional antigens and restrict the development and function of so-called innate or pre-set T cells. The CD1 proteins present lipids to invariant natural killer T (iNKT) cells; [20], and MR1 proteins present vitamin metabolites to activate or inhibit mucosal-associated invariant T (MAIT) cells [21].

Among nonmammalian species, multiple *MHC-Ib* gene families following various and complex evolutionary patterns have been found in the genome of many jawed vertebrates’ species, whereas jawless vertebrates (agnathans) have neither *MHC-I* nor *MHC-II* genes (reviewed in [3]). In elasmobranchs (cartilaginous fish, such as sharks and rays), extensive searches have revealed the occurrence of a large diversity of *MHC-Ib* gene lineages besides *MHC-Ia*, pointing to the early emergence of *MHC-Ib* diversification in jawed vertebrates’ evolution [22,23,24]. The diversity of these *MHC-Ib* genes varies significantly across taxa, grouping in distinct gene lineages with complex evolutionary histories [3]. Notably, sharks have two families of expanded *MHC-Ibs* genes, the *UBA* [25] and *UEA* [22]). The *UBA* family is derived from the *UAA* family, while the *UEA* lineage is more ancient. There are also additional single-copy *MHC-Ib* genes (*UDA*, *UFA*, and *UGA*).

In bony fish, a variety of MHC-Ib gene lineages have been described based on evolutionary relationships. These genes have been grouped into distinct lineages (*U*-, *Z/ZE*-, *L*- and *S*-lineage) and are differentially distributed among species [26]. For example, genes belonging to the *U*-lineage (containing both putative *MHC-Ia* and *MHC-Ib* genes) are broadly represented among divergent species, whereas to date, the *L*-lineage that consists of highly divergent *MHC-Ib* genes has only been identified in salmonids and cyprinids [27]. Interestingly, the salmonid *L*- lineage genes, including *Sasa-LIA* and *Sasa-LGA1*, are differentially induced in response to microbial challenges [28,29]. Expansion of *MHC-Ib* gene families also occurred in multiple other fish species [30]. Genes of the *MHC-Ib U*-lineage have been detected in Gadiformes (an order of ray-finned fishes including cod, pollock, haddock, etc.), in which several species have around 100 gene copies. Species within *Percomorphaceae* (percomorph fishes; a large grouping of spiny-finned teleost) have up to 80 copies. Within Gadiformes, high copy numbers above 40 were observed in as many as 12 species. The Atlantic cod, *Gadus morhua* (family *Gadidae*), has a peculiar immune system, characterized by the loss of the MHC-II pathway, and an extreme expansion of the *MHC-Ib* gene repertoire [31,32]. Many of these *MHC-Ib* genes are expressed in cod tissues [33]. In addition, a group of *G. morhua* MHC-Ib genes colocalize with tapasin on endolysosomes, suggesting that peptide-loading assistance and stabilization of MHC-Ib molecules can occur outside the endoplasmic reticulum, a feature reminiscent of CD1 antigen presentation [34]. It is tempting to speculate that these MHC-Ib molecules localized in endocytic vesicles can present peptides derived from extracellular pathogens and thus, may represent an ancestral cross-presentation pathway antecedent to the MHC-II pathway [35]. The loss of the MHC-II pathway has been found in many species of the *Gadidae* family and in the pipefish *Syngnathus typhle*, from the distantly related *Syngnathidae* fish family [36]. However, it appears that in bony fish, *MHC-I* gene expansions have occurred multiple times independently of the loss of the MHC-II pathway [30]. As such, there are still many unresolved aspects of the evolution and specialized functions of these *MHC-Ib* lineages [37].

In amphibians, there appears to be little evidence of wide *MHC-Ib* gene expansion outside the *Xenopus* genus/lineage (subfamily *Xenopoidae*) discussed in detail in Section 2.3. Nevertheless, there is evidence of *MHC-Ib* gene occurrence, and *Ranidae* (Ranid frogs) have as many as 20 MHC-Ib genes [38]. While there is only a single *MHC-Ia* gene per genome in *Xenopus* species, *MHC-Ia* gene copies vary in number among other anuran species (one to five). The recent progress in genome sequencing should soon provide reliable information about *MHC-I* genes in salamanders to elucidate the putative expansions of *MHC-I* genes in these species [39].

While the immune system of reptiles is still understudied, the information from the increasing number of genome sequences available should help to fill this gap in our knowledge. Notably, a large number of *MHC-I* loci of unclear types have been reported in multiple reptile species [40]. In addition, *CD1* gene homologs have been reported (see Section 2.1. below).

In birds, new sequencing technology has revealed greater *MHC* variation than previously expected. For example, several *MHC-Ib* lineages have been found in the red-billed gull (*Larus scopulinus*) and sparrow species (Passer) [41,42]. In chicken, an additional *MHC-like* region (the *Rfp-Y* system) contains *MHC-I* genes distinct from the classical B locus where the *MHC-Ia* gene is located. Some of these genes appear to be nonpolymorphic, suggesting that they encode MHC-Ib molecules [43,44].

The immunological significance of expanded *MHC-Ib* gene lineages distinct from *MHC-Ia* genes found in genomes across all jawed vertebrates remains mostly unclear and understudied. However, the fact that many of these genes have not been deleted or silenced suggests a selection for some specialized function. While the phylogeny of the different *MHC-Ib* lineages remains complex and elusive, some insights have been gathered regarding the two *MHC-Ib* gene families *CD1* and *MR1* that in humans and mice present lipids and vitamin derivatives, respectively.

### 2.1. CD1

*CD1* is a nonpolymorphic *MHC-Ib* gene located on a chromosome distinct from the MHC locus. Although CD1 associates with the β2-microglobulin-like MHC-Ia molecules, it is targeted to endosomal vesicles akin to MHC-II molecules [20]. *CD1* gene homologs have been identified in both reptiles and birds but not in bony fishes or amphibians [45].

In birds, chickens express at least one *CD1*-like gene (*chCD1.1*), which shares structural similarities with mammalian CD1d, hinting at the possible existence of iNKT-like cells in birds [46,47]. In mammals, *CD1* underwent further diversification by gene duplication, resulting in multiple isoforms—such as *CD1a–e* in humans—which led to specialized adaptation to bind various lipid antigens and to interact with different T cell subsets [48,49,50]. *CD1* gene homologs have also been described in the green anole lizard and crocodilians [51].

Recently, two *CD1*-like MHC-linked, single- or low-copy *MHC-Ib* genes called *UFA* and *UGA* have been identified in sharks and rays (Flajnik pers. Communication). Crystal structure and mass spectrometry analyses suggest that these molecules bind lipids comparable to those bound by human CD1d. This discovery implies that while the *CD1* gene was lost in bony fish and amphibians, it likely emerged nearly 500 million years ago in early jawed vertebrates in parallel with *MHC-Ia* and *MHC-II* genes. Interestingly, while no *CD1* gene homolog is found in the amphibian *Xenopus*, a monogenic *MHC-Ib* lineage, *XNC10*, exhibits multiple analogous features suggesting a convergent *CD1*-like function (see below *Xenopus* MHC-Ib).

Another interesting finding regarding the origin of the molecular domain binding to phospholipids, is the gene for the endothelial protein C receptor (*PROCR*) that encodes the endothelial cell protein C receptor (EPCR). The *PROCR* gene shares 20% sequence identity with genes of the CD1 family of molecules, and the two encoded molecules share some structural features related to phospholipid binding [52]. Phylogenetic studies using conserved “phylogenetic marker motifs” suggest that the *MHC-Ib* lineages *CD1/PROCR* and *UT* (Marsupials) were established before the emergence of tetrapod species [45]. The finding of shared molecular domains binding to phospholipids via molecules other than *MHC-Ibs* raises the possibility of an evolutionary origin of lipid binding preceding peptide binding in immune defense.

### 2.2. MR1

The MR1 gene encodes an MHC-Ib molecule in mice and humans that bind and present vitamin B_2_ derivatives to MAIT cells. *MR1* gene homologs appear evolutionarily more recent than *CD1*, as they are confined to mammals [53,54]. No *MR1* genes have been identified in fish and amphibians [48,55]. While no *MR1* gene homolog can be identified in most reptiles’ genomes, turtles may be an exception since an *MR1*-like gene has been reported, although its encoded molecule lacks the ability to bind riboflavin metabolites effectively [45]. In chicken, an MHC-Ib molecule called YF1*7.1 may represent an avian *MR1*-like gene. It shares 38% sequence identity with human *MR1* and is structurally similar to it [56].

These gene candidates in turtles and chickens suggest that the precursor mechanisms for the MAIT system might have emerged outside mammals, even though true MAIT cells themselves did not emerge until the mammals evolved. In addition, *MR1* genes are absent in monotreme mammals like the platypus [48], and emerged first in marsupial and eutherian (placental) mammals [57]. Furthermore, the *CD1* gene has been independently lost (or pseudogenized) in several placental mammals, including carnivores (cats, dogs, and pandas), xenarthrans like the armadillo, and lagomorphs (rabbits and relatives) [45,58]. This loss coincides with the loss of a particular invariant V segment of the *TCRα* (*TRAV1*; [55]. Conversely, *TRAV1* is present in all the species that have a functional *MR1* gene [58]. This suggests that the MR1 gene is not only highly conserved across mammalian species but also undergoing purifying selection via restriction of the semi-invariant *TCR TRAV1* [48]. This is further supported by the fact that MR1 molecules from one species can often activate MAIT cells from another species (e.g., bats), highlighting an important and stable function that has tolerated little structural variation [59,60].

### 2.3. Xenopus MHC-Ib (uba) Genes

To date, *Xenopus* is the only species outside mammals in which *MHC-Ib* genes have been extensively studied at the genomic, molecular, and functional levels. In contrast to a single polymorphic *MHC-Ia* gene residing in the MHC locus per haploid genome, the *Xenopus* genome displays a large number of *MHC-Ib* genes [61]. In both *X. tropicalis* and its sister species *X. laevis* that diverged some 40 million years ago, there is an extended family of *MHC-Ib* genes clustered in the telomeric region of chromosome 8, far outside the proper MHC locus. It is tempting to speculate that this localization outside the MHC locus far from the centromere may protect *MHC-Ia* from gene conversion and/or allow for extensive recombination and rapid diversification.

Three additional *MHC-Ib* genes have also been identified in different locations of the *X. tropicalis* and *X. laevis* genomes [62]. Many of these *MHC-Ib* genes are also present in other *Xenopus* species of the *Xenopodinae* subfamily [63,64]. We have renamed these genes following the nomenclature system for ectothermic vertebrates *MHC* genes [65]. The classical MHC-Ia chain is referred as “*uaa*”, and is sequentially followed by other families or groups: *uba*, *uca*, *uda*, *uea*, etc. Thus, for *Xenopus*, “*uba*” stands for *MHC -I (u)*, family or group (*b*), and alpha chain (*a*) [66]. There are at least 23 *MHC-Ib* genes or *mhc1-uba* loci (formerly named *XNCs* genes) in *X. laevis* and 29 genes in *X. tropicalis*, plus several pseudogenes indicative of past gene duplications. These genes are heterogeneous, oligomorphic, and less ubiquitously expressed than the single polymorphic *MHC-Ia* gene [5,64,67,68]. Moreover, several of these *MHC-Ib* genes (*XNC 1*, *4*, *9*, *10*, *11*, and *14*) in both *X. tropicalis* and *X. laevis* are preferentially expressed in the thymus by radiosensitive thymocytes early in development, rather than by thymic stroma or epithelium as is the case for MHC-Ia [64,69]. This expression pattern is consistent with a role in iT cell differentiation. Among *MHC-Ib* genes, *XNC10* and *XNC4* in *X. laevis* have been functionally characterized (summarized in Figure 1 and Figure 2).

*-XNC10* (*mhc1-uba10.1.L*) is a monogenic, highly conserved gene lineage present in all ten species of the *Xenopodinae* subfamily [63,64]. It is expressed early in ontogeny (on thymocytes in both larval and adult frogs) and convergent indirect evidence suggests multiple features analogous to mammalian CD1d [64,69]. Although XNC10’s ligands have not yet been identified, several lines of evidence are consistent with XNC10 representing a CD1 functional analog accommodating lipid antigens, namely its expression pattern, the invariant T cell subset that it restricts, its roles in antiviral and tumor immunity, and even 3D structural modeling, all which point toward a CD1d-like lipid-presenting function. XNC10 restricts an invariant Vα6-Jα1.43 T cell lineage, which can be considered as an “iNKT-like” subset in frogs. XNC10-tetramers produced in insect cell lines have shown that this iT cell population represents a substantial (5–10%) fraction of circulating and splenic T cells in both tadpoles and adult frogs [70]. As detailed in Section 5.1, XNC10 appears to have a dual role in *Xenopus* immunity; it governs the development of Vα6 iT cells and also delivers an inhibitory “checkpoint” signal to cells that express it [71,72].

*-XNC4* (*mhc1-uba4.L*) is a gene encoding a distinct MHC-Ib molecule in *X. laevis* and *X. tropicalis*. XNC4 molecules exhibit some features analogous to human HLA-F, potentially binding to unusually long peptide ligands of 12 to 16 amino acids (Adams and Robert, unpublished). So far, no clear anchor, strong motifs or overrepresented peptides have been found. Similarly, sequence alignment analysis of the XNC4 putative binding domain with *MHC-Ia*, *MR1* or *CD1d* did not reveal obvious conservation of amino acid residues involved in terminal peptide anchoring at either ends of the antigen-binding groove of the two shallow pockets, A and F, although the XNC4 putative F pocket is hydrophobic [63]. Based on reverse genetics, XNC4 interacts with an iT cell subset defined by the Vα45-Jα1.14 TCR rearrangement (discussed in more details in Section 5.2).

Several other *XNC* genes exist in *Xenopus*, each likely selecting a distinct iT cell subset or performing a specialized role. Notably, aside from the spleen, *XNC14* expression is largely restricted to the intestine, in both tadpoles and adult frogs (Robert unpublished). Indeed, transcriptomic studies suggest that infection dramatically upregulates certain *XNC* genes in *Xenopus*, although full details remain under investigation (Table 1).

## 3. Preset or Innate-like T Cells (iT Cells)

Conventional αβT cells are defined by the vast TCR repertoire that they express, which is generated during their differentiation within the thymus via RAG-mediated somatic recombination and selection by MHC-Ia and MHC-II molecules expressed on thymic stroma. In contrast, iT cells such as mammalian iNKT and MAIT undergo a distinct differentiation program that includes positive selection by thymocytes instead of cortical epithelial cells [75]. In mammals, the hallmark of preset iαβ T cells is the expression of the transcription factor PLZF (encoded by the *Zbtb16* gene), that induces a tissue-residency differentiation program [76]. Induction of PLZF is linked to selection by CD4/CD8 double-positive (DP) thymocytes [76,77]. This selection step depends on homotypic SLAM-dependent interactions and on TCR signals delivered by ligands on DP thymocytes [78]. As such, these cells acquire pre-activated effector functions during thymic development. In both humans and mice, iNKT and MAIT cells already acquire memory-like effector functions during development, making them ready to promptly secrete cytokines upon activation [55].

### 3.1. iNKT Cells

In mice and humans, Type I iNKT cells express a semi-invariant TCR, with a fixed TCRα chain (*Vα14-Jα18* in mice, *Vα24-Jα18* in humans) paired with limited TCRβ chains (reviewed in [20]). iNKT cells can recognize glycolipid antigens, both self-derived and microbial, presented by the MHC-Ib molecule CD1d [49,50]. CD1d is expressed on cortical thymocytes and antigen-presenting cells; it presents self or foreign lipids (e.g., the marine sponge glycolipid α-Galactosylceramide [α-GalCer], microbial glycolipids, and self-sphingolipids) to iNKT. Positive selection of iNKT cells requires CD1d expression on DP thymocytes [75]. Differentiated iNKT cells possess characteristics of both T cells and NK cells, as they often express NK lineage markers and can rapidly secrete large amounts of cytokines (IFN-γ, IL-4, etc.) upon activation [79]. In the periphery, stressed or infected cells can upregulate CD1d and present altered self-lipids, thereby triggering iNKT responses. Although iNKT cells make up a small fraction of T cells in the blood in humans (about 0.1–1%), their higher clonal frequency compared to conventional T cells allows them to provide a rapid and significant response without requiring extensive proliferation [80,81]. Type II NKT cells that are also lipid-specific CD1d-restricted T cells but express other less invariant TCR and generally do not recognize α-galactosylceramide (α-GalCer), as well as NKT cell interacting with other CD1 molecules CD1a, etc., are not considered here [82]. CD1d-restricted iNKT cells perform pleiotropic functions in different tissues by secreting a vast array of pro-inflammatory and cytotoxic molecules. In cancer, iNKT cells play a dual role: they can directly kill certain tumor cells and secrete cytokines that activate other immune cells, thus bridging innate and adaptive immunity in cancer surveillance [83,84].

Among the different iNKT cell effector functions characterized, anti-tumor activity and control of inflammation are of particular interest here. A large body of data indicate that in humans, iNKT cells promote anti-tumor immunity through rapid IFN-γ production and cytotoxicity [85,86,87,88]. In mouse models, activation of iNKT cells with α-GalCer significantly enhances tumor rejection: α-GalCer-pulsed dendritic cells can eradicate established melanoma metastases. Conversely, mice lacking iNKT cells develop spontaneous or chemically induced tumors more readily [89,90,91]. iNKT cells kill CD1d^+^ tumor cells directly in a CD1d-dependent, Fas/FasL-mediated fashion and indirectly by stimulating NK and CD8 T cells via IFN-γ to attack tumors [85,92]. iNKT cells can detect self-lipid antigens generated within the tumor lipidome, which are presented by CD1d (reviewed in [83]). Since, unlike MHC-Ia, CD1d is nonpolymorphic, it is an attractive target for tumor immunotherapy using engineered iNKT cells. As such, chimeric antigen receptor (CAR) technology has been applied to iNKT cells to enhance their anti-tumor activity, taking advantage of their innate-like recognition properties and tumor-homing capabilities. Unlike conventional CAR-T cells, CAR-iNKT cells retain their ability to recognize lipid antigens via CD1d, allowing them to simultaneously engage both CAR-targeted and CD1d-restricted tumor pathways. This dual-targeting potential improves tumor clearance and reduces the risk of antigen escape, making CAR-iNKT therapy a promising strategy in cancer immunotherapy [93].

iNKT cells can also regulate inflammation. In type-2 inflammation such as allergy, asthma, or helminth infection, the iNKT2 cell subset contributes to airway inflammation by producing IL-4 and IL-13 [94]. In mouse models, iNKT-derived IL-13 can promote mucus production and IgE class switching, whereas IFN-γ from iNKT can suppress Th2 pathology. Because iNKT cells respond to self-lipids and microbial glycolipids, they can rapidly amplify immune responses to stress or infection, while at the same time, they can also dampen responses via IL-10-secreting regulatory NKT subsets [85]. In sepsis and autoimmunity models, iNKT-derived IFN-γ and IL-4 play opposing “double-edged” roles in controlling excessive inflammation. Thus, iNKT cells act as sensors of cellular stress (via CD1d) and modulators of cytokine balance, linking innate detection to adaptive immunity [84,95]. The regulatory potential of iNKT cells is also underscored by the IL-10-mediated immunoregulatory role of intestinal iNKT cells in controlling the pathogenic functions of mucosal T helper subsets and in maintaining intestinal immune homeostasis [96].

### 3.2. MAIT Cells

MAIT cells are another semi-invariant innate αβ T cell subset that uses a conserved TCRα chain (*Vα7.2-Jα33* in the *TRAV1-2* gene family in humans) and recognizes metabolites derived from the riboflavin (vitamin B_2_) biosynthesis pathway [20,21,97]. These metabolites, such as 5-(2-oxopropylideneamino)-6-D-ribitylaminouracil (5-OP-RU), are presented by MR1 [21]. MAIT cells are abundant in tissues such as the liver and mucosal surfaces in humans and respond to bacteria and yeast that produce riboflavin. They can detect infected cells (e.g., epithelial cells, macrophages, dendritic cells) that are expressing MR1 molecules loaded with microbial riboflavin metabolites. MAIT cells then actively secrete cytokines like IFN-γ and IL-17 as well as cytotoxic molecules to lyse these infected cells. MAIT cells appear to be restricted to mammals since MR1 is absent in birds, reptiles, and amphibians, and thus no true MAIT cells are present in these species [58]. The emergence of MR1 in early mammals (~170 million years ago) led to the development of a conserved MAIT TCR-MR1 system with antimicrobial activity, suggesting that this system conferred a significant evolutionary advantage to mammals [45,59]. During MAIT cell development, MR1 is required at each intrathymic stage [98]. In adults, many cells in various tissue constitutively express MR1 at low levels; this provides an effective survey for microbial riboflavin metabolites. When an infected or malignant cell presents an MR1-bound ligand, MAIT cells rapidly activate.

MAIT cells play dual roles in cancer and inflammation [99]. By recognizing MR1-presented metabolites, MAIT cells can directly target MR1-expressing tumor cells presenting vitamin-B-derived metabolites [100]. They show direct cytotoxicity against K562 cells via degranulation of granzyme B and perforin. MAIT-derived IFN-γ has been shown to enhance NK cell anti-tumor activity, and MAIT cells can infiltrate tumors and promote pro-inflammatory microenvironments (e.g., IFN-γ and TNF-α production). However, MAIT cells often accumulate in tumors where they may produce TNF-α, IL-17, or IL-13-cytokines known to enhance tumor growth and suppress cytotoxic immunity. For instance, IL-17^+^ MAIT cells in breast and lung tumors correlate with poor prognosis, and MAIT cells in prostate cancer can express PD-L1 that inhibits αβ T cells. The net effect of MAIT cells in cancer may depend on the context: in some settings (e.g., glioma, colorectal cancer) high MAIT cell numbers or MR1 expression correlates with better outcomes, while in other settings (e.g., cervical, lung cancer) MAIT cells are exhausted or suppressive. In summary, MAIT cells can act as “first responders” at mucosal sites to detect early malignant changes. Their presence in tissues like the colon and cervix has been suggested to reflect early defense against transformation. However, chronic tumor-associated inflammation may shift MAIT cells toward a pro-tumor phenotype [101].

## 4. Human iT Cells During Early Development

iT cells, including iNKT and MAIT cells, are present early in human life but are not fully mature at birth. iNKT cells emerge during fetal thymic development, with detectable populations in fetal tissues such as the liver and intestine before birth. Cord blood studies show that neonatal iNKT cells are mostly naive (CD4^+^, CD45RA^+^) with low cytotoxic capacity, but are poised to expand after exposure to environmental antigens [102]. Postnatally, iNKT cells gradually acquire effector functions as they encounter glycolipid antigens presented by CD1d on antigen-presenting cells [103]. Even in low numbers, neonatal iNKT cells can rapidly produce IFN-γ and IL-4 upon stimulation, suggesting a capacity to influence early immune responses, particularly during inflammatory challenges or infection [104,105].

MAIT cells are rare in human cord blood and only slowly expand in the first years of life, in parallel with microbiota colonization. In neonates, MAIT cells (Vα7.2^+^ CD161^hi^) are largely naive and functionally immature, with poor cytokine production compared to adult cells [106]. However, small populations of MR1-restricted T cells may contribute to early-life mucosal defense, and their gradual expansion suggests that invariant T cells are part of the immune system’s adaptation to the postnatal environment [107]. Overall, invariant T cells in newborns provide an early, innate-like layer of immunity before conventional adaptive immunity reaches full maturity. They may help regulate inflammation during this critical period and protect against early-life infections, while their developmental maturation ensures a growing role in immune surveillance as infants encounter environmental microbes and stressors.

## 5. Xenopus MHC-Ib/iT Immune Surveillance System

Like all anurans, *Xenopus* tadpoles undergo radical tissue remodeling during metamorphosis (e.g., tail resorption, organ reorganization [108]). This major transformation includes the immune system (reviewed in [109]). Notably, during this transition, immune tolerance to self-antigens is crucial. *Xenopus* tadpoles naturally do not express MHC-Ia on their tissues until metamorphosis [68,110]. During the larval stage, the expression of MHC-Ia molecules and components of the antigen-processing machinery, such as immunoproteasome subunits, is minimal [68]. This feature is thought to prevent autoimmunity against larval or neoantigens. As metamorphosis proceeds, MHC-Ia is upregulated [111]. In this context, MHC-Ib (XNC) molecules play a critical role: XNC10 and other XNCs are expressed on larval thymocytes from early stages and are required for the emergence of several iT cell subsets before metamorphosis [64,69]. These iT cells are central to controlling microbial infections and malignancies in the absence or with minimal contribution from conventional T cells. Thus, while some mammalian MHC-Ib molecules protect embryonic tissues in utero (e.g., HLA-G), amphibian MHC-Ib molecules protect larval tissues pre-metamorphosis. Both systems use MHC-Ib to enforce self-tolerance during vulnerable developmental windows.

As a result, the T cell repertoire in tadpoles differs significantly from that in adult frogs. Tadpoles predominantly generate invariant TCRs with minimal junctional diversity, with a focus on a few rearrangements shared by many T cells [70]. Deep sequencing of tadpole T cells revealed that just six unique invariant TCRα sequences account for over 80% of the TCRα chains in the larval T cell pool, particularly among CD8^−^ or CD8^+lo^ T cells [70]. This indicates that the larval immune system is dominated by a small number of preset or iT cell subsets. After metamorphosis, adult frogs shift to producing a much more diverse TCR repertoire, and MHC-Ia-restricted T cells become more prominent.

### 5.1. XNC10-Restricted iVα6 T Cells

T cells expressing an invariant TCR α chain (*Vα6-Jα1.43*) strictly depend on the MHC-Ib XNC10 for their development and function [70]. When *XNC10* is knocked down or knocked out (by RNA interference or CRISPR mutagenesis), iVα6 T cells fail to develop, making the tadpoles more susceptible to viral infections [67,112]. Moreover, the absence of XNC10 significantly impairs the ability of both tadpoles and adult frogs to defend against the ranavirus Frog Virus 3 (FV3). iVα6 T cells are quick to respond to FV3 infection and accumulate at the infection sites as early as six hours. Furthermore, the transient deletion of iVα6 T cells by injecting XNC10 tetramers dramatically compromises viral clearance and the survival of infected tadpoles [113].

Interestingly, iVα6 T cells and XNC10 also play a critical role in tumor immunity. *X. laevis* is one of the few nonmammalian species with transplantable tumors that have been characterized. Several lymphoid tumor cell lines were derived from spontaneous thymic tumors and adapted to cell culture conditions, tested for their molecular signatures and immune-related responses in vitro and in vivo [114,115,116]. As a comparative model to investigate the role of MHC-Ib and iT cells in tumor immunity, these *X. laevis* tumor cell lines are valuable. When iVα6 T cells are temporarily depleted using XNC10-tetramers, tadpoles show a reduced ability to reject transplanted thymic lymphoid tumors that express multiple *XNC* genes including *XNC10* [117]. Silencing *XNC10* transcripts in these tumors results in their rejection, which suggests that XNC10 overexpression is used as an evasion mechanism [71,118]. Intriguingly, XNC10-deficient transgenic tadpoles that also lack Vα6 iT cells were resistant to these lymphoid tumors, uncovering a potential new function of XNC10 besides Vα6 iT cell development [117]. Collectively, these data indicate that the XNC10-restricted iVα6 T cell subset is a key component of tumor surveillance in tadpoles. The development and function of XNC10-restricted iVα6 T cells are summarized in Figure 1.

### 5.2. XNC4 Interacting iVα45 T Cells

The invariant *iVα45-Jα1.14 TCRα* rearrangement is one of the six over-dominant rearrangements identified via RACE-PCR and deep sequencing of splenic T cells in tadpoles [70]. T cells expressing this invariant *iVα45-Jα1.14 TCR* were subsequently found to require the MHC-Ib XNC4 for their development [73]. Genetic ablation of *XNC4* abolishes this iVα45 T cell population and dramatically compromises antibacterial immunity. XNC4-deficient tadpoles show greatly increased mortality upon *Mycobacterium marinum* infection [73]. Similarly, both *Jα1.14* knockout by CRISPR/Cas9 and shRNA silencing of the *Vα45-Jα1.14 TCR* transcripts dramatically impair tadpole resistance to *M. marinum*. By contrast, these iVa45-deficent animals retain resistance to ranavirus FV3 infection comparable to wild-type control tadpoles, indicating that the XNC4-Vα45 axis is specialized for mycobacterial defense [73]. Together, these findings imply that XNC4-interacting iV45 T cells provide a dedicated antibacterial/regulatory arm of immunity, akin to the roles of certain MHC-Ib molecules in mammalian inflammation and tolerance [73,74]. This also suggests a division of roles among invariant T cell subsets in tadpoles with one dedicated to antiviral responses and another to antibacterial defense. This functional specialization in amphibians highlights how iT cells may have evolved to address specific immune needs. Development and function of XNC4 and interacting iVα45 T cells are summarized in Figure 2.

## 6. Evolutionary Perspective of the MHC-Ib/iT Immune Surveillance System in Jawed Vertebrates

Both mammalian and *Xenopus MHC-Ib* genes have low allelic diversity and tissue- restricted expression. However, sequence homology is low between species and *Xenopus MHC-Ib* genes have no clear direct orthologs in mammals. For example, *XNC10* has no mammalian counterparts. Remarkably, *Xenopus* species (e.g., *X. laevis* vs. *X. tropicalis*) share high sequence similarity in *XNC* domains (up to ~95%) despite 45 million years of divergence, which is an unusual conservation for *MHC-Ib* genes. In mammals, *MHC-Ib* genes have often evolved rapidly and differ even across mouse strains. In contrast, *XNC* genes (especially *XNC10*) are conserved across many *Xenopus* species and genome ploidies. However, some *XNCs* mirror mammalian *MHC-Ib* functions. The *XNC10* gene is required for a population of iT cells analogous to CD1d-restricted iNKT. Thus, although the sequences differ, *Xenopus* may have evolved an MHC-Ib and iT cell system that is functionally convergent with the CD1d/iNKT axis in mammals. Similarly, XNC4 presents some similarity with HLA-F, which in humans binds to unusually long peptides and regulates antimicrobial immune response particularly against *M. tuberculosis* as well as certain autoimmune processes [16]. XNC4 also appears to bind to unusually long peptides and is critical for controlling the infection of nontuberculous mycobacteria such as *M. marinum* in tadpoles. Whether XNC4 can signal through NK cell receptors in the same way as HLA-F is currently unknown.

Across jawed vertebrates, iT cells may provide early-life immune defenses particularly adapted for non-placental animals that are not protected by the maternal environment and thus are under pressure to develop a functional immune system very early during ontogeny. While there is a large body of data on *MHC-Ib* genes from genome and transcriptome studies, little is known to date about putative iT cells or even overrepresentation of particular *abTCR* rearrangements outside mammals and *Xenopus*. We recently embarked on mining a RNAseq dataset comparing the transcriptomes of two spadefoot toad species (*Scaphiopus couchii* and *Pelobates cultripes*) for *TCRα* rearrangements (A. Savage and J. Robert, unpublished). Preliminary evidence suggests that some *TCRα* transcripts are indeed overrepresented in both species.

The *Xenopus* model offers compelling evidence that even without MHC-Ia peptide presentation, an organism can still effectively defend against tumors and infections by relying on iT cells. With the rapid progress in omics technologies and the use of reverse genetics via the CRISPR/Cas9 system now applicable to comparative immunology, a better insight into the potential function of the different *MHC-Ib* family members will be forthcoming. Notably, better characterization of T cells with limited *TCR* repertoires that interact with some of these MHC-Ib across ectothermic vertebrates, will permit us to evaluate how important the MHC-Ib/iT cells system has been in the evolution of the vertebrate immune system. Table 2 provides an overview of the current knowledge of MCH-Ib and iT cells across jawed vertebrates.

## Figures and Tables

**Figure 1 cells-14-01592-f001:**
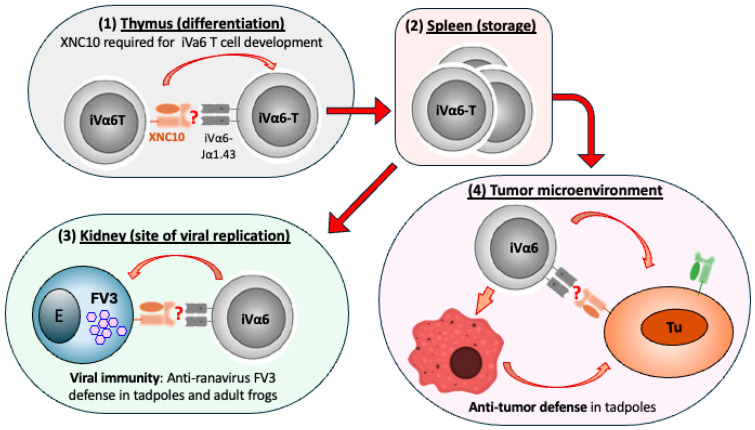
**Model of XNC10-restricted iVα6 T cell development and function.** (**1**) In the thymus, thymocytes that have rearranged Vα6 with Jα1.43 can bind to XNC10 expressed on other thymocytes, receiving a signal to differentiate and exit the thymus. (**2**) iVα6 T cells migrate into the spleen where they reside. (**3**) Upon an undefined signal, iVα6 T cells are rapidly recruited to the site of viral infection (kidney) to promote XNC10-dependent antiviral responses. (**4**) iVα6 T cells are also recruited to the site of a lymphoid tumor where they promote an anti-tumor immune response either directly, by interacting with XNC10 expressed on tumor cells and/or indirectly, by acting on macrophages. High levels of XNC10 (and other XNCs) contribute to immune evasion of lymphoid tumors. (?) XNC10 ligands have not been identified but are postulated to be lipids.

**Figure 2 cells-14-01592-f002:**
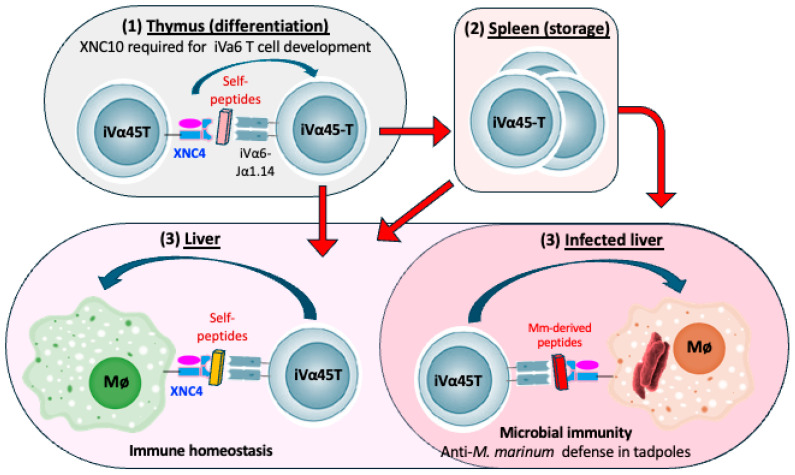
**Model of XNC4-mediated development and function of iVα45 T cells.** (**1**) In the thymus, thymocytes that have rearranged Vα45 with Jα1.43 can bind to XNC4 (which possibly presents self-peptides expressed on other thymocytes) to receive a signal to differentiate and exit the thymus. (**2**,**3**) iVα45 T cells migrate into the spleen and the liver, where they contribute to immune homeostasis. (**4**) Upon *M. marinum* infection, more iVα45 T cells are recruited in the liver where they potentiate the antimicrobial activity of infected macrophages, possibly expressing XNC4 binding to *M. marinum*-derived peptides.

**Table 1 cells-14-01592-t001:** Summary of current information about mhc1-uba (XNC) genes.

*Mhc1* ^1^*-uba*	*X. tropicalis* (Expression)	*X. laevis* (Expression) ^2^	Putative Peptide Binding Domain and Ligands ^3^	Function (Known or Putative)	Ref.
**1**	*mhc1-uba1.1-1.2*	***mhc1-uba1.L*** (Thymocytes, Sol., skin, Intestine, from early dev.)	F pocket hydrophobic	Putative role during larval development	[67,68]
**2**	*mhc1-uba2.1-2.2*	*mhc1-uba2.S*	?	?	
**3**	*mhc1-uba3*	*mhc1-uba3.L*	F pocket hydrophobic	?	
**4**	*mhc1-uba4* (Thymocytes)	***mhc1-uba4*** (Thymocytes., spleen, liver, intestine, Infect. sites)	F pocket hydrophobic, bind unusually long peptides	Anti-mycobacterial immunity	[73,74]
**5**	*mhc1-uba5*	*mhc1-uba5.L*	F pocket hydrophobic	?	
**6**	*mhc1-uba6.1-6.2*	*mhc1-uba6.1-6.3L*, *mhc1b-uba6.4S*		?	[61,68]
**7**	*mhc1-uba7.1-7.6 (Ubiquitous)*	*mhc1-uba7.L* (Ubiquitous including thymic stroma)	F pocket partially hydrophobic	?	[68]
**8**	*unidentified*	*mhc1-uba8.1-8.4L* (lungs mainly)	?	?	
**9**	*mhc1-uba9*	*mhc1-uba9.L*	?	?	
**10**	*mhc1-uba10* (Thymocytes)	***mhc1-uba10.1*** (Thymocytes from onset of organogenesis, spleen, infect. sites, lymphoid tumors)	Open F pocket, potential for binding lipids	Anti-viral (FV3) immunity Anti-lymphoid tumor immunity	[69]
**11**	*mhc1-uba11* (Thymocytes)	*mhc1-uba11.L* (Thymocytes, Spleen, lymphoid tumors)	?	Cancer biology	[71]
**12**	*mhc1-uba12*	*unidentified*	?	?	
**13**	*mhc1-uba13.1-13.5*	*mhc1-uba13.1*, *mhc1b-uba13.5.L*	?	?	
**14**	*mhc1-uba14*	***mhc1-uba14* ^4^**	F pocket hydrophobic	Mucosal immunity	
**16**	*mhc1-uba16.1-16.4*	*unidentified*	?	?	
**17**	*mhc1-uba17*	*unidentified*	F pocket hydrophobic	?	

^1^ The genes identification and nomenclature use the version 10.1 of genome assemblies for both *X. tropicalis* and *X. laevis* as described in [66]. ^2^
*L* and *S* refer to the long and small chromosome of *X. laevis* [66], and bold characters indicate genes that have been studied to some extent in *X. laevis*. ^3^ [63]. ^4^ Robert, unpublished.

**Table 2 cells-14-01592-t002:** Overview of occurrence of MHC-Ib and iT cells during jawed vertebrate evolution.

Vertebrate Taxa.	MR1	CD1	MHC-Ib ^4^	MAIT	iNKT	abTCR Diversity	abTCR Limited	Notable Functional Features
Cartilaginous fish (sharks, rays)	−	CD1-like	Expanded	−	?	+	?	Lipid binding on CD1-like
Bony fish	−	−	Expanded	−	?	+	?	CD1-like endosomal processing in Atlantic cod
Amphibians *Xenopus*	−	−	Expanded	−	iVa6T	+	+	Antiviral and antitumor activity of XNC10-restricted iVa6 T cells Antimicrobial and regulatory activity of XNC4/iVa45 T cells
*Other frogs*	−	−	Expanded	−	?	+	?	
*Salamander*	−	−	Expanded	−	?	+	?	
Reptiles	− ^1^	+	Expanded	?	?	+	?	No functional data to date
Birds Chickens	? (+) ^2^	+ +	+ +	? +	+	+	+	Mammalian CD1d tetramers bind to the chicken T cell subset
Marsupials	+	+	Expanded	+	?	+	?	No functional data to date
Euterian	+ ^3^	+	+	+	+	+	+	Anti-tumor and regulatory iNKT cells
Humans	+	a−e	+	+	+	+	+	Antimicrobial and regulatory MAIT cells

^1^ Turtles may have an MR1-like gene. ^2^ Chicken YF1*7.1 shares significant sequence identity and structural similarity with MR1. ^3^ Loss in carnivores (cats, dogs, and pandas), armadillos, and lagomorphs. ^4^ Numbers in parenthesis are the maximum reported. +, −, and ? indicate the presence, absence, or unknown occurrence, respectively, of the feature in the phylogenetic class.

## Data Availability

Not applicable.

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
