# Peer review of "Evolutionary Perspective of Nonclassical MHC Class I and Innate-like T Cells Relevance in Immune Surveillance"

_cells, 2025, doi:10.3390/cells14201592_

Round 1
Reviewer 1 Report
Comments and Suggestions for Authors
This is an informative and well written review of the evolution of innate T cells and class Ib molecules. I only have a few questions/comments.
--PROCR is not mentioned (that I found) in the review. It seems to be CD1-related, as suggested by Dijkstra in the IMC review on class I, and it has interesting properties. I think it should be mentioned.
--It is interesting that the rfpy and xnc genes are moved out of the MHC proper to a telomeric region of the same chromosome. This might be to protect class Ia from gene conversion and/or allow for rapid diversification being far from the centromere to allow for extensive resombination.
--Does xnc4 have a hydrophilic binding site, suggesting that it binds peptides? And, does it lack the residues that lock in peptides, i.e. is it more like class II?
--Has a survey been done on the xnc families, to suggest which ones bind peptides and which have hydrophobic binding sites (predicted)? Would it be worth having a table on the XNC characteristics (binding site, expression (overall and thymic epithelium vs thymocytes), conservation in amphibia (or at least Xenopus species)?
--It’s been noted that the monomorphic bony fish Z and shark UDA have hydrophilic binding sites and might bind peptides, like HLA-E and Qa1. They also have an affinity in the phylogenetic trees, which is interesting.
--Shars have two families of expanded class Ibs, the UBA (Bartl, JI, 1997, 159:6097) and UEA (Almeida, referenced). UBA is derived from UAA while UEA is more ancient. They also have the single-copy UDA, UFA, and UGA. Saying that sharks have 100s of class Ib genes is overstating (Table 1).
--Since class Ia and class Ib go back to the beginning, any idea on which might have come first, a peptide or lipid binder?
Author Response
1) PROCR is not mentioned (that I found) in the review. It seems to be CD1-related, as suggested by Dijkstra in the IMC review on class I, and it has interesting properties. I think it should be mentioned.
Answer: Thank you for the suggestion. We added th following paragraph:
Another interesting finding regarding the origin of the molecular binding domain of phospholipids, is the gene for the endothelial protein C receptor (PROCR) that encodes the endothelial cell protein C receptor (EPCR). The PROCR gene shares 20% sequence identity with genes of the CD1 family of molecules and the two encoded molecules share some structural features related to phospholipid binding [50]. Phylogenetic studies using conserved “phylogenetic marker motifs” suggest that the MHC-Ib lineages CD1/PROCR and UT (Marsupials) were established before the emergence of tetrapod species [43].
2) It is interesting that the rfpy and xnc genes are moved out of the MHC proper to a telomeric region of the same chromosome. This might be to protect class Ia from gene conversion and/or allow for rapid diversification being far from the centromere to allow for extensive recombination.
Answer: Thank you for this suggestion. We added: “It is tempting to speculate that this localization outside the MHC locus far from the centromere may protect class Ia from gene conversion and/or allow for extensive recombination and rapid diversification.”
3) Does xnc4 have a hydrophilic binding site, suggesting that it binds peptides? And, does it lack the residues that lock in peptides, i.e. is it more like class II?
Answer: We still have to study XCN4 structure in more details. We added a sentence: “So far, no clear anchor, strong motifs or overrepresented peptides have been found.”
4) Has a survey been done on the xnc families, to suggest which ones bind peptides and which have hydrophobic binding sites (predicted)? Would it be worth having a table on the XNC characteristics (binding site, expression (overall and thymic epithelium vs thymocytes), conservation in amphibia (or at least Xenopus species)?
Answer: We feel we don’t have sufficient information for a table at this time.
5) It’s been noted that the monomorphic bony fish Z and shark UDA have hydrophilic binding sites and might bind peptides, like HLA-E and Qa1. They also have an affinity in the phylogenetic trees, which is interesting.
Answer: Thank you for this information. But we have decided to omit this because we do not discuss HLA-E and Qa1much in this review.
6) Sharks have two families of expanded class Ibs, the UBA (Bartl, JI, 1997, 159:6097) and UEA (Almeida, referenced). UBA is derived from UAA while UEA is more ancient. They also have the single-copy UDA, UFA, and UGA. Saying that sharks have 100s of class Ib genes is overstating (Table 1).
Answer: Thank you for the precision. We have deleted numbers in table 1, and added the following:
“Notably, sharks have two families of expanded MHC-Ibs genes, the UBA [24] and UEA [21]). The UBA family is derived from the UAA family, while the UEA lineage is more ancient. There also additional single-copy MHC-Ib genes (UDA, UFA, and UGA). “
7) Since class Ia and class Ib go back to the beginning, any idea on which might have come first, a peptide or lipid binder?
Answer: This is interesting question but perhaps beyond the scope to this review.
Reviewer 2 Report
Comments and Suggestions for Authors
Overview: In general, this is a well-written and comprehensive review. I especially liked the first paragraph of section 5 explaining the importance of innate T cells ( iT cell subsets) during tadpole development. However, there are some sections of the text that are redundant and should be revised. That is, some text should be deleted or greatly condensed acknowledging the information will be presented in later sections. Without knowing to what extent each manuscript receives copy-editing attention, I have noted many grammatical errors that should be corrected before this manuscript is published. They are listed in some detail. While these issues may seem minor, the smooth flow of words is critical to understanding the major points of the review.
Specific Suggestions or Questions
- As presented in Figs. 1 and 2, the suggestion of iT cell “education” by other radio-sensitive developing thymocytes in the amphibian thymus is new to me. Is this already published and accepted in the literature?
- Lines 165-166. You need a reference to the study showing that chicken T cells can bind to mammalian CD1d tetramers.
- Line 205. Somewhere in section 2.3, it might be good to explain the history of the names for the MHC-Ib genes. Explain briefly where the "uba" terminology comes from.
- Lines 212-213. Do you mean to say "in a different location" or in "different locations"? That is, are the three additional MHC-1b genes together in one location or scattered?
- Line 377. When describing the functions of MAIT cells, please explain whether they specifically target host cells infected with bacteria or yeast. Are they essentially primed sentinels to destroy infected host cells?
- Lines 458-484. Some of this information about XNC10 and iVα6 T cells is redundant to what is presented as a subsection in Section 2.3. You should decide where it fits best. It might be better to omit from section 2.3 and present it more fully in section 5.1.
- Lines 486-504. This information is redundant to what is previously described in section 2.3. I suggest you abbreviate what is mentioned in section 2.3 and keep this information here as section 5.2.
Minor correections
- Line 29 and elsewhere. Throughout the MS, please delete double parentheses.
- Line 32. Change "jaw vertebrates" to "jawed vertebrates".
- Line 35. Add comma after "the focus of this review".
- Line 51. Add the word "on" after the word "focused".
- Line 59 and elsewhere. Please indicate which publications are reviews and which are primary literature.
- Line 60. Change "jaw' to "jawed".
- Line 61. Change "include" to "includes".
- Line 63. Add the word "of" after the word "fragments".
- Line 64 and 81. Change letter "b" to Greek symbol "beta".
- Line 66. Change "is well known" to "is a well-known".
- Line 71. Add the word "genes" after "MHC-Ib".
- Line 83. Add the word "proteins" after "Other MHC-Ib". Make it clear that you refer to the expressed proteins and not the genes.
- Line 86. Change "gene" to "genes".
- Lines 88-89. The phrase beginning with "notably UT8" is unclear. Do you mean to say "notably UT8 is expressed by developing thymocytes."
- Line 92. Add the word "proteins" after the words "CD1 family”. Here again, please make clear when you are referring to the expressed proteins instead of the genes.
- Line 93. The word "cells" appears to be redundant in this sentence. There should also be a comma placed before "MR1" to connect the two parts of the sentence.
- Line 96. Change "vertebrates" to "vertebrate". In this sentence, "vertebrate" is an adjective describing species.
- Lines 105-107. This is a very awkward and unclear sentence. Break up the sentence and change to "In bony fish, a variety of MHC-Ib lineages have been described based on evolutionary relationships. They have been grouped into distinct lineages (U-, Z/ZE-, L- and S-lineage) and are differentially distributed among species (Lukacs, Harstad et al. 2010).”
- Line 118. Is morhua a species of fish? If so, please say so and give the common name or the full genus and species name.
- Line 120. Add a comma after the word "reticulum".
- Line 123. Add the word "they" after the words "and thus,".
- Line 129. Change "There is" to "There are".
- Line 131. Change "In amphibian" to "In amphibians".
- Line 136. Change "provide soon" to "soon provide".
- Line 140. Change "sequence" to "sequences".
- Line 144. Change "where" to "in which".
- Lines 155-156. Change "has" to "have". Change "lipid ad vitamin derivative" to "lipids and vitamin derivatives, respectively".
- Line 159. Add "the" after the word "Although".
- Line 169. Change "subset" to "subsets".
- Line 177. Change "amphibian" to "amphibians".
- Line 179. Change "exhibit" to "exhibits".
- Lines 183-184. Change "appear evolutionary more recent" to "appear to be evolutionarily more recent".
- Line 185. Change "find in fish and amphibian" to "found in fish and amphibians".
- Lines 186-188. This sentence is very poorly written: "While no MR1 can be identified in most reptile’s genome, turtles may be an exception since an MR1-like gene has reported, although it encoded molecule lacks the ability to bind riboflavin metabolites effectively”. Please change to “While no MR1 can be identified in most reptile’s genomes, turtles may be an exception since an MR1-like gene has been reported, although its encoded molecule lacks the ability to bind riboflavin metabolites effectively.”
- Line 202. Is TCR TRAV11 correct or should it be TCR TRAV1?
- Lines 206-207. This sentence is badly written. I suggest: "To date Xenopus is the only species outside mammals in which the MHC-Ib genes have been studied at the genomic, molecular, and functional levels."
- Line 208. Change "display" to "displays a".
- Line 213. Please add the word "genes" after "MHC-Ib". Are they detected in the genome or as proteins? Please be clear.
- Line 229. Change "thymocyte" to "thymocytes" and add "a" before the word "signal".
- Line 231. Change "response" to "responses".
- Lines 232-234. Please revise this sentence: "iV6 T cells are also recruited the site of lymphoid tumor where they promote tumor immune response either directly by integrating with XNC10 expressed by tumor cells and/or indirectly by acting on macrophage.” I suggest the following: iV6 T cells are also recruited to the site of a lymphoid tumor where they promote tumor immune responses either directly by integrating with XNC10 expressed by tumor cells and/or indirectly by acting on macrophages.
- Line 239. Replace "thymocyte" with "thymocytes".
- Line 250. Do you mean "role in antiviral and tumor immunity"?
- Lines 238-240. Lines 238-240. This is an unclear sentence. Please revise. "This his selection step depends on homotypic SLAM-dependent interactions and TCR signals delivered by ligands on DP thymocytes requires SLAM-dependent interactions and TCR triggering by ligands presented on DP thymocytes.”
- Line 301. Delete the word "a" following the word "acquire".
- Line 311. Add "is" after "CD1d".
- Line 319. Change "human" to "humans" and change "convT" to "conventional T cells".
- Line 323. Define what alpha-GalCer is.
- Line 330. Change "function" to "functions".
- Line 332. Change "human" to "humans".
- Line 347. Did you mean "anti-tumor" instead of "anti-important" activity?
- Line 372. Change "human" to "humans" and change "recognize" to "recognizes".
- Line 433. It looks like you forgot to cite a review about the immune system reorganization at metamorphosis.
- Line 444. Change "T cell" to "T cells".
- Lines 474-475. Change "have revealed to be" to "are".
- Lines 477-478. Chang "tumor" to "tumors". Change "evasion mechanism" to "an evasion mechanism".
- Line 483. Insert the word "the" before "XNC10-restricted". Insert the word "in" before the word "tadpoles".
- Line 486. The heading should be section 5.3.
- Lines 487-488. Change "6" to "six". Change "rearrangement" to "rearrangements".
- Line 490. Change "to requires" to "to require".
- Line 507. Change "jaw" to "jawed".
- Line 519. Change "present" to "presents".
- Line 519-521. Change "present" to "presents". Change "human" to "humans". Change "regulate" to "regulates". Change "response" to "responses". tuberculosis should be in italics.
- Line 525-526. Change "jaw" to "jawed". Add the word "the" before "maternal”. Add the word "are" after the words "environment and".
- Lines 529-530. Change "its" to "their". Add "an" before "RNAseq dataset".
- Lines 537-538. Add a comma after "comparative immunology". Change "MHC-Ib family is warranted." to "MHC-Ib family members will be forthcoming."
- Lined 539-542. Change "T cell" to "T cells". Change "repertoire to "repertoires". Add "us" between the words "permit" and "to". Change "Table provide" to "Table 1 provides".
Comments on the Quality of English Language
There are many minor grammatical errors in the use of written English. This does not diminish the importance of the science, but it can be distracting. Thus, I have made many suggestions to improve the flow of the text.
Author Response
Specific Suggestions or Questions
1) As presented in Figs. 1 and 2, the suggestion of iT cell “education” by other radio-sensitive developing thymocytes in the amphibian thymus is new to me. Is this already published and accepted in the literature?
Answer: Yes, this has been published and the work is cited.
2) Lines 165-166. You need a reference to the study showing that chicken T cells can bind to mammalian CD1d tetramers.
Answer: We have deleted this statement because we cannot find references.
3) Line 205. Somewhere in section 2.3, it might be good to explain the history of the names for the MHC-Ib genes. Explain briefly where the "uba" terminology comes from. 
Answer: We added in the section 2.3 the following: “We have renamed these genes following the nomenclature system for ectothermic vertebrates MHC genes (Ballingall et al., 2018). The classical MHC-Ia chain is referred as “uaa”, and is sequentially followed by other families or groups: uba, uca, uda, uea, etc. Thus, for Xenopus “uba” stands for MHC -I (u), family or group (b), alpha chain (a)."
4) Lines 212-213. Do you mean to say, "in a different location" or in "different locations"? That is, are the three additional MHC-1b genes together in one location or scattered?
Answer: different locations. Thank you for this.
5) Line 377. When describing the functions of MAIT cells, please explain whether they specifically target host cells infected with bacteria or yeast. Are they essentially primed sentinels to destroy infected host cells?
Answer: We added a sentence: “They can detect host cells (e.g., epithelial cells, macrophages, dendritic cells) that are presenting MR1 loaded with microbial riboflavin metabolites.”
6) Lines 458-484. Some of this information about XNC10 and iVα6 T cells is redundant to what is presented as a subsection in Section 2.3. You should decide where it fits best. It might be better to omit from section 2.3 and present it more fully in section 5.1.
Answer: We deleted the redundant information in section 2.3.
7) Lines 486-504. This information is redundant to what is previously described in section 2.3. I suggest you abbreviate what is mentioned in section 2.3 and keep this information here as section 5.2.
Answer: We deleted the redundant information in section 2.3. and revised the section 5.2 as follow:
“Genetic ablation of XNC4 abolishes this iVα45 population and dramatically compromises antibacterial immunity. XNC4-deficient tadpoles show greatly increased mortality upon Mycobacterium marinum infection (Edholm, Banach et al. 2018). Similarly, both Jα1.14 knockout by CRISPR/Cas9 and shRNA silencing of the Vα45-Jα1.14 TCR transcripts dramatically impair tadpole resistance to M. marinum. By contrast, these iVa45-deficent animals retain resistance to ranavirus FV3 infection comparable to wild type control tadpoles, indicating that the XNC4-Vα45 axis is specialized for mycobacterial defense. Together, these findings imply that XNC4-interacting iV45 T cells provide a dedicated antibacterial/regulatory arm of immunity, akin to the roles of certain MHC-Ib molecules in mammalian inflammation and tolerance (Edholm, Banach et al. 2018, Rhoo, Edholm et al. 2019). This also suggests a division of roles among invariant T cell subsets in tadpoles with one dedicated to antiviral responses and another to antibacterial defense (Edholm et al. 2018). This functional specialization in amphibians highlights how innate T cells may have evolved to address specific immune needs. Development and function of XNC4 and interacting iVα45 T cells are summarized in Fig. 2.”
Minor corrections
Line 29 and elsewhere. Throughout the MS, please delete double parentheses.
Answer: We have formatted the references.
Line 32. Change "jaw vertebrates" to "jawed vertebrates". 
Line 35. Add comma after "the focus of this review". 
Line 51. Add the word "on" after the word "focused". 
Line 59 and elsewhere. Please indicate which publications are reviews and which are primary literature.
Answer: Done accordingly.
Line 60. Change "jaw' to "jawed". 
Line 61. Change "include" to "includes". 
Line 63. Add the word "of" after the word "fragments". 
Line 64 and 81. Change letter "b" to Greek symbol "beta". 
Line 66. Change "is well known" to "is a well-known". 
Line 71. Add the word "genes" after "MHC-Ib". 
Line 83. Add the word "proteins" after "Other MHC-Ib". Make it clear that you refer to the expressed proteins and not the genes. 
Line 86. Change "gene" to "genes". 
Lines 88-89. The phrase beginning with "notably UT8" is unclear. Do you mean to say "notably UT8 is expressed by developing thymocytes."
Answer: Yes. We have clarified this.
Line 92. Add the word "proteins" after the words "CD1 family”. Here again, please make clear when you are referring to the expressed proteins instead of the genes. 
Line 93. The word "cells" appears to be redundant in this sentence. There should also be a comma placed before "MR1" to connect the two parts of the sentence. 
Line 96. Change "vertebrates" to "vertebrate". In this sentence, "vertebrate" is an adjective describing species. 
Lines 105-107. This is a very awkward and unclear sentence. Break up the sentence and change to "In bony fish, a variety of MHC-Ib lineages have been described based on evolutionary relationships. They have been grouped into distinct lineages (U-, Z/ZE-, L- and S-lineage) and are differentially distributed among species (Lukacs, Harstad et al. 2010).” 
Line 118. Is morhua a species of fish? If so, please say so and give the common name or the full genus and species name. 
Line 120. Add a comma after the word "reticulum". 
Line 123. Add the word "they" after the words "and thus,". 
Line 129. Change "There is" to "There are". 
Line 131. Change "In amphibian" to "In amphibians". 
Line 136. Change "provide soon" to "soon provide". 
Line 140. Change "sequence" to "sequences". 
Line 144. Change "where" to "in which". 
Lines 155-156. Change "has" to "have". Change "lipid ad vitamin derivative" to "lipids and vitamin derivatives, respectively". 
Line 159. Add "the" after the word "Although". 
Line 169. Change "subset" to "subsets". 
Line 177. Change "amphibian" to "amphibians". 
Line 179. Change "exhibit" to "exhibits". 
Lines 183-184. Change "appear evolutionary more recent" to "appear to be evolutionarily more recent". 
Line 185. Change "find in fish and amphibian" to "found in fish and amphibians".
Lines 186-188. This sentence is very poorly written: "While no MR1 can be identified in most reptile’s genome, turtles may be an exception since an MR1-like gene has reported, although it encoded molecule lacks the ability to bind riboflavin metabolites effectively”. Please change to “While no MR1 can be identified in most reptile’s genomes, turtles may be an exception since an MR1-like gene has been reported, although its encoded molecule lacks the ability to bind riboflavin metabolites effectively.”
Answer: We changed the sentence as suggested. Thank you.
Line 202. Is TCR TRAV11 correct, or should it be TCR TRAV1?
Answer: Yes, TRAV1. Corrected. Thank you.
Lines 206-207. This sentence is badly written. I suggest: "To date Xenopus is the only species outside mammals in which the MHC-Ib genes have been studied at the genomic, molecular, and functional levels."
Answer: We changed the sentence as suggested. Thank you.
Line 208. Change "display" to "displays a". 
Line 213. Please add the word "genes" after "MHC-Ib". Are they detected in the genome or as proteins? Please be clear. 
Line 229. Change "thymocyte" to "thymocytes" and add "a" before the word "signal". 
Line 231. Change "response" to "responses". 
Lines 232-234. Please revise this sentence: "iV6 T cells are also recruited the site of lymphoid tumor where they promote tumor immune response either directly by integrating with XNC10 expressed by tumor cells and/or indirectly by acting on macrophage.” I suggest the following: iV6 T cells are also recruited to the site of a lymphoid tumor where they promote tumor immune responses either directly by integrating with XNC10 expressed by tumor cells and/or indirectly by acting on macrophages.
Answer: We changed the sentence as suggested. Thank you.
Line 239. Replace "thymocyte" with "thymocytes". ! 
Line 250. Do you mean "role in antiviral and tumor immunity"? text modification
Answer: We revised as follow:
“Although XNC10’s ligands have not yet been identified, several lines of evidence are consistent with XNC10 accommodating lipid antigens: its expression pattern, the in-variant T-cell subset it restricts, its roles in antiviral and tumor immunity, and even 3D structural modeling, all point toward a CD1d-like lipid-presenting function. XNC10 re-stricts an invariant Vα6-Jα1.43 T-cell lineage, essentially an “iNKT-like” subset in frogs.”
Lines 298-300. This is an unclear sentence. Please revise. "This his selection step depends on homotypic SLAM-dependent interactions and TCR signals delivered by ligands on DP thymocytes requires SLAM-dependent interactions and TCR triggering by ligands presented on DP thymocytes.” 
Answer: We corrected: “This selection step depends on homotypic SLAM-dependent interactions and on TCR signals delivered by ligands on DP thymocytes.”
Line 301. Delete the word "a" following the word "acquire". 
Line 311. Add "is" after "CD1d".
Line 318-319. Change "human" to "humans" and change "convT" to "conventional T cells".
Line 343. Define what alpha-GalCer is.
Answer: We added the marine sponge glycolipid Galactosylceramide [alpha-GalCer]…
Line 330. Change "function" to "functions".
Line 332. Change "human" to "humans".
Line 347. Did you mean "anti-tumor" instead of "anti-important" activity? 
Line 372. Change "human" to "humans" and change "recognize" to "recognizes". 
Line 453. It looks like you forgot to cite a review about the immune system reorganization at metamorphosis.
Answer: We added the reference.
Line 444. Change "T cell" to "T cells".
Lines 474-475. Change "have revealed to be" to "are". 
Lines 477-478. Chang "tumor" to "tumors". Change "evasion mechanism" to "an evasion mechanism".
Line 483. Insert the word "the" before "XNC10-restricted". Insert the word "in" before the word "tadpoles".
Line 486. The heading should be section 5.3.5.2 
Lines 487-488. Change "6" to "six". Change "rearrangement" to "rearrangements".
Line 490. Change "to requires" to "to require".
Line 527. Change "jaw" to "jawed".
Line 519. Change "present" to "presents".
Line 519-521. Change "present" to "presents". Change "human" to "humans". Change "regulate" to "regulates". Change "response" to "responses". tuberculosis should be in italics. 
Line 525-526. Change "jaw" to "jawed". Add the word "the" before "maternal”. Add the word "are" after the words "environment and".
Lines 539-530. Change "its" to "their". Add “an" before "RNAseq dataset".
Lines 537-538. Add a comma after "comparative immunology". Change "MHC-Ib family is warranted." to "MHC-Ib family members will be forthcoming." 
Line 539-542. Change "T cell" to "T cells". Change "repertoire” to "repertoires". Add "us" between the words "permit" and "to". Change "Table provide" to "Table 1 provides".
Reviewer 3 Report
Comments and Suggestions for Authors
An example is that the title suggests to be a review on nonclassical MHC whereas it is on nonclassical MHC class I.
- "At the cell surface MHC-Ia presents peptide fragments 7 to 8 amino acids" (most peptides are 9 aa, 7 may not occur naturally)
- "In birds, MHC-I U-lineage genes have been detected in Gadiformes, where several 144 species had around 100 gene copies, followed by species within Percomorphaceae with up 145 to 80 copies. Within Gadiformes, high copy numbers above 40 were observed in as many 146 as 12 species. In chickens an additional MHC-like region (the Rfp-Y system) contains 147 MHC-I genes distinct from the classical B locus containing MHC-Ia." The authors here seem to mistake Gadiformes for birds (chicken belong to "Galliformes") and describe the data for fish species instead of for birds.
Comments on the Quality of English LanguagePlease use any software for screening for such errors. You will see that those errors are far beyond the occasional small error.
Author Response
I added MHC class I in the title and corrected 8 to 9 amino acid. The 3rd mistake was already corrected in the last submitted version.
Round 2
Reviewer 2 Report
Comments and Suggestions for Authors
Thank you for making the changes. I have no more suggestions for changes
Author Response
Comment 1: provide more details on XCN4 structure.
Answer: we added the following sentence: So far, no clear anchor, strong motifs or overrepresented peptides have been found. Similarly, sequence alignment analysis of the XNC4 putative binding domain with MHC-Ia, MR1 or CD1d did not reveal obvious conservation of amino acid residues involved in terminal peptide anchoring at either ends of the antigen-binding groove of the two shallow pockets, A and F, although the XNC4 putative F pocket is hydrophobic [63].
Comment 2: please insert a table.
Answer: We have inserted a new table (table 1). Summary of current information about mhc1-uba (XNC) genes.
Comment 3: since class Ia and class Ib go back to the beginning, any idea on which might have come first, a peptide or lipid binder? please add a sentence to answer this question.
Answer: we added the following sentence at the end of paragraph 2.1: The finding of shared molecular domains binding phospholipids by molecules other than MHC-Ibs raises the possibility of an evolutionary origin of lipid binding preceding peptide binding in immune defense.